# EXPLAIN, AGREE AND LEARN: A RECIPE FOR SCALABLE NEURAL-SYMBOLIC LEARNING

## ABSTRACT

Recent progress in neural-symbolic AI (NeSy) has demonstrated that neural networks can benefit greatly from an integration with symbolic reasoning methods in terms of interpretability, data-efficiency and generalisation performance. Unfortunately, the symbolic component can lead to intractable computations for more complicated domains. This computational bottleneck has prevented the successful application of NeSy to more practical problems. We present **EXPLAIN**, **AGREE** and **LEARN**, an alternative paradigm that addresses the scalability problem of probabilistic NeSy learning. **EXPLAIN** leverages sampling to obtain a representative set of possible explanations for the symbolic component driven by a newly introduced diversity criterion. Then **AGREE** assigns importance to the sampled explanations based on the neural predictions. This defines the learning objective, which for sufficiently many samples is guaranteed to coincide with the objective used by exact probabilistic NeSy approaches. Using this objective, **LEARN** updates the neural component with direct supervision on its outputs, without the need to propagate the gradient through the symbolic component. Our approximate paradigm and its theoretical guarantees are experimentally evaluated and shown to be competitive with existing exact probabilistic NeSy frameworks, while outperforming them in terms of speed.

## 1 INTRODUCTION

The field of neural-symbolic AI (NeSy) aims to combine the perceptive capabilities of neural networks with the reasoning capabilities of symbolic systems. NeSy methods tend to generalize better and require less training data compared to purely neural methods. However, NeSy methods have to tackle the problem of propagating the learning signal through the symbolic component in order to train the neural network. They usually resort to performing this propagation exactly, which is often intractable for more complex systems and leads to poor scalability.

In this paper, we introduce the EXPLAIN, AGREE, and LEARN (EXAL) paradigm, which aims to address scalability issues in probabilistic NeSy learning. In the EXPLAIN step, explanations at the level of the neural output are sampled for the data labels. The AGREE step bounds the likelihood using these explanations. The LEARN step then trains the neural network to optimize this bound. We present an implementation of the EXAL paradigm for probabilistic NeSy, with an EXPLAIN procedure aiming to sample a *diverse* set of explanations. We develop a basic sampling procedure and a learnable extension that explicitly maximizes diversity using GFlowNets. Furthermore, we establish the correctness of the EXAL paradigm, the importance of every step, and provide theoretical bounds on the error of the learning objective. Moreover, the increased diversity of our solution provides better bounds on the learning objective compared to uniform sampling. Finally, we show through an ablation study on MNIST digits that the three stages are all essential to guarantee proper learning. The EXAL paradigm can be applied to general probabilistic NeSy frameworks, but the details of the EXPLAIN algorithm depend on the nature of the symbolic component. We will from now on focus on a probabilistic logic symbolic component.

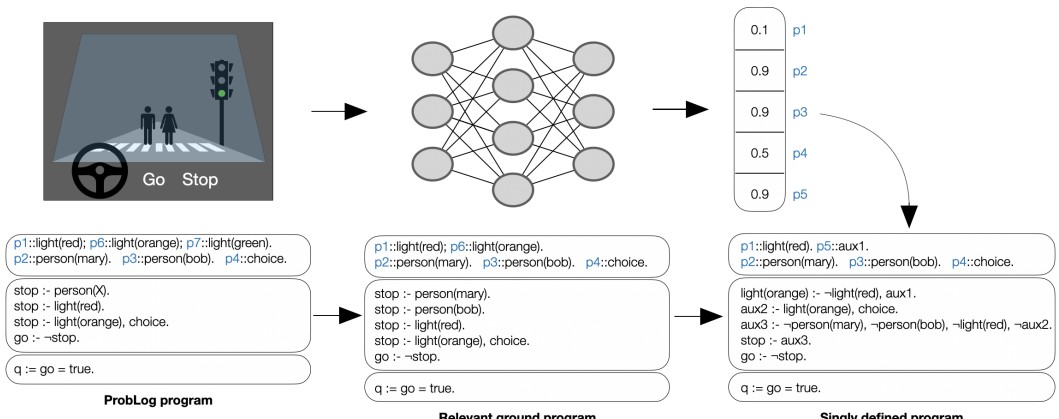

Figure 1: **Traffic light example**. A logic program determines whether the car should go or stop depending on the color of the traffic light and whether people are walking on the road or not, as sensed by a camera installed on the car (see top-left image). Indeed, we have a ProbLog program expressed in first-order logic (bottom-left), a ground program relevant for the query (bottom) and the equivalent program used by our inference strategy (bottom-right). The neural network (top) transform the sensed image into a vector of probabilities on the facts of the logic program (top-right).

## 2 BACKGROUND

**Probabilistic Logic Programming.** A *logic program* is a set $\mathcal{R}$ of *rules*, i.e. expressions of the form $a \leftarrow b_1 \wedge ... \wedge b_n$, where $a$ is the *head*, and the $b_i$ are *body literals*. Rules are to be interpreted as follows: IF all the body literals are true THEN the head is true. Rules with an empty body are *facts*. In first-order logic programs, literals can have (variable) arguments. First-order logic programs can be grounded into propositional logic by substituting all variables with grounded terms.

ProbLog (De Raedt et al., 2007) extends logic programs to *probabilistic logic programs* through the introduction of probabilistic facts. A probabilistic fact is of the form $p_i :: f_i$ where $f_i$ is a logical fact and $p_i$ is the probability of $f_i$ being true. In ProbLog, each probabilistic fact $f_i$ corresponds to an independent Boolean random variable that is true with probability $p_i$. Let $\mathcal{H}$ denote the Herbrand base of the program, i.e. the set of all atoms, and $\mathcal{F} \subseteq \mathcal{H}$ the set of facts. Given a subset $F \subseteq \mathcal{F}$ of facts assigned the value true, the probability of $F$ is defined as $P(F) = \prod_{f_i \in F} p_i \prod_{f_i \notin F} (1 - p_i)$. Additionally, a probabilistic logic program defines the probability of a query $q$, given by $P(q) = \mathbb{E}_F[\mathbb{1}_{F \models_{\mathcal{R}} q}]$, with $\mathbb{1}$ an indicator and $F \models_{\mathcal{R}} q$ denoting whether $q$ is logically entailed by the rules $\mathcal{R}$ if only the facts in $F$ are true.

## 3 EXPLAIN, AGREE AND LEARN: THE RECIPE

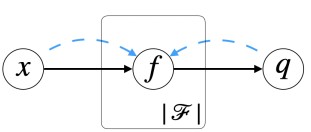

Figure 2: Probabilistic graphical model formulation of NeSy learning.

NeSy learning is framed in Bayesian terms using the probabilistic graphical model in Figure 2. It contains a sub-symbolic input data vector $x \in \mathbb{R}^D$, the query $q$, and an assignment[1] $f$, which is defined below:

**Definition 1.** A *(partial) assignment* under a rule set $\mathcal{R}$ of a program is a mapping $\nu : H \to \{0, 1\}$ with $H \subseteq \mathcal{H}$ such that none of the rules in $\mathcal{R}$ are violated when every atom $a$ in $H$ gets assigned the value $\nu(a)$. An assignment is *complete* if $\text{dom}(\nu) = \mathcal{H}$. A *completion* of a partial assignment $\nu$ is a complete assignment $\mu$ such that $\forall a \in \text{dom}(\nu) : \mu(a) = \nu(a)$. We will later use the notation $\nu + l$ for a literal $l$ to denote a new assignment

---

[1]In an abuse of notation, we will use $f$ both to denote the mapping as well as the logic program obtained by assigning the truth value of $f$ to every fact $f_i \in \mathcal{F}$

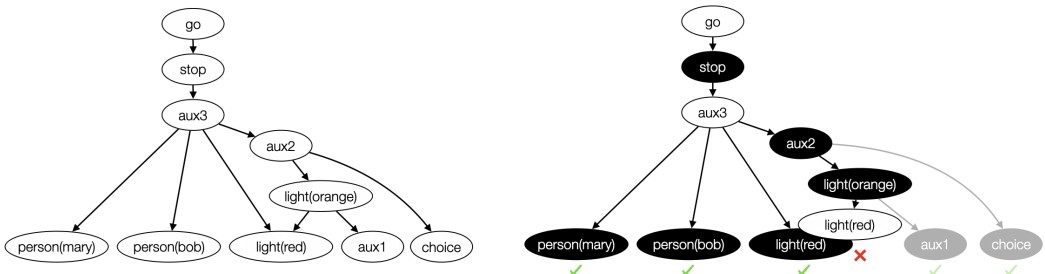

Figure 3: On the left, we have the DAG induced by the logic program from the traffic light example. On the right, we have an example of the search subgraph obtained by the sampling procedure. White and black colours are used to indicate whether a ground atom is assigned a true or false value, respectively. Additionally, black nodes require to randomly select which atom from their body must be assigned a value, thus being called choice nodes. Leaf nodes (nodes without outgoing arrows) are probabilistic facts. In the example on the right, we observe that the partial assignment of the probabilistic facts leads to a conflict. The program admits two valid partial assignments, namely $\{\neg person(mary), \neg person(bob), \neg light(red), \neg aux1\}$ and $\{\neg person(mary), \neg person(bob), \neg light(red), \neg choice\}$.

that maps all atoms in $dom(\nu)$ to the same value as $\nu$ and maps the atom in $l$ to 1 if $l$ is a positive literal and to 0 otherwise.

The logic component is $P(q \mid f) = \mathbb{1}_{f \models_{\mathcal{R}} q}$ and the probabilistic perception component $P(f \mid x)$ is modeled by a neural network $g : \mathbb{R}^D \to [0,1]^{|\mathcal{F}|}$ as $P(f \mid x) = \prod_i g_i(x)^{f_i}(1 - g_i(x))^{1-f_i}$. Additionally, we define the auxiliary distribution $Q(f \mid x, q) = w_{x,f} \mathbb{1}_{f \in \Omega_q}$, where $\Omega_q = \{f \mid f \models_{\mathcal{R}} q\}$ is the set of assignments that make $q$ true, i.e. the models of $q$. $\Omega_q$ is the *support* of $Q$, whereas $w_{x,f} \in \mathbb{R}^+$ are the positively-defined *weights* of $Q$, such that $\sum_{f \in \Omega_q} w_{x,f} = 1$. The objective function is a lower bound of the log-likelihood over a training data set of samples $\mathcal{D} = \{(x_j, q_j)\}_{j=1}^N$:

$$\log P(\mathcal{D}) \propto \sum_{j=1}^N \log P(q_j \mid x_j) \geq \sum_{j=1}^N \sum_{f \in \Omega_{q_j}} Q(f \mid x_j, q_j) \log \frac{P(f \mid x_j)}{Q(f \mid x_j, q_j)} = -\sum_{j=1}^N \mathrm{KL}(Q_j | P_j). \tag{1}$$

For the full derivation we refer to App. A. Maximizing the lower bound in Eq. 1 consists of minimizing the Kullback-Leibler (KL) term through standard gradient descent. The KL bound is made tighter by updating the weight function $w_{x,f}$. Notably, the training signal for the neural component $P(f \mid x)$ is provided by the auxiliary distribution $Q$, which sidesteps the traditional requirement of backpropagating gradients through the logic program (Manhaeve et al., 2018; Xu et al., 2018; Ahmed et al., 2022). In other words, we avoid expensive exact probabilistic inference based on weighted model counting or knowledge compilation (Oztok & Darwiche, 2015) during learning. While the sidestepping of this requirement is appealing, it is important to mention that the support $\Omega_q$ of $Q$ is not given. In the following sections, we are going to provide the ingredients of our recipe to estimate $\Omega_q$, to learn the weight function $w_{x,f}$ and finally learn the neural network.

### 3.1 EXPLAIN: SAMPLE EXPLANATIONS

First, we will show how to interpret a ProbLog program as a directed acyclic graph (DAG). Then, we will introduce the notion of an explanation and give the EXPLAIN algorithm to sample explanations for a query. EXPLAIN uses the DAG structure to guide its search. Finally, we define the diversity of sampled explanations and devise a learning strategy to improve the diversity of EXPLAIN's output.

**Logic program to DAG.** We will assume the following properties of the logic program: *ground*, *singly defined*, i.e., every atom can occur in the head of at most one rule, and *acyclic*, i.e., there exists a total ordering on the atoms such that the head of every rule is greater than all the atoms in the body. A program that is not ground or singly defined can always be transformed into a program that does have these properties. For grounding a program, we refer to (Gebser et al., 2007), while App. B gives

---

**Algorithm 1** Sampling algorithm EXPLAIN$(\phi, q, \nu, \leq_q, u)$

---

1: **Input:** logic formula $\phi$, query $q$, (partial) assignment $\nu$, partial ordering $\leq_q$, value function $u$.
2: **Output:** explanation for the query encoded in $\phi$, or $\emptyset$ if a conflict is encountered.
3:
4: $\phi \leftarrow$ unit propagate$(\phi)$
5: **if** $\phi$ is empty **or** $\nu$ is an explanation for $q$ **then**
6:     **return** $\nu$
7: **end if**
8: **if** $\phi$ has an unsatisfiable clause **then**
9:     **return** $\emptyset$
10: **end if**
11: $c \leftarrow$ smallest clause in $\phi$ according to $\leq_q$
12: $l \leftarrow$ random literal in $c$ chosen with weights $u$
13: $\nu \leftarrow \nu + l$
14: **return** EXPLAIN$(\phi \wedge l, q, \nu, \leq_q, u)$

---

a transformation to make any program singly defined. As an example, Figure 1 shows a ProbLog program that is transformed into its ground singly defined equivalent. Additionally, the acyclicity property ensures that we can interpret a ground singly defined program as a directed acyclic graph $\mathcal{G} = (\mathcal{H}, \mathcal{L})$, where $\mathcal{H}$ is the Herbrand base and $\mathcal{L}$ is the set of links determined by the program rules. Specifically, $\mathcal{L}$ contains an edge from the head of a rule to every atom in the body. In other words, $\mathcal{G}$ is the Prolog dependency graph with reversed edges. The DAG obtained from the traffic light ProbLog program is given in Figure 3.

**Parsimonious sampling.** We introduce some definitions useful for the EXPLAIN algorithm.

**Definition 2.** Given an assignment $\nu$, an atom $a \in \text{dom}(\nu)$ is *explained* in $\nu$ if $a$ is a fact or if $a$ is the head of a rule $a \leftarrow b_1, ..., b_n$ and the value of $a$ is entailed by the values of $b_i$ given by $\nu$.

**Definition 3.** An *explanation* for a query $q$ is an assignment $\nu$ with $\nu(q) = 1$ such that every atom in $\text{dom}(\nu)$ is explained.

**Definition 4.** A *value function* $u : \mathcal{L} \to \mathbb{R}^+$ assigns a positive real value to every edge in $\mathcal{G}$.

For now we assume that $\forall l \in \mathcal{L} : u(l) = 1$. We will later see how to generalize/parameterize the value function and learn it.

The EXPLAIN algorithm is used to sample an explanation for a query $q$. It is a probabilistic extension of the Davis–Putnam–Logemann–Loveland (DPLL) algorithm (Davis & Putnam, 1960; Davis et al., 1962). Because DPLL works on logical formulas, the ground program first has to be encoded as a formula $\phi$ using Clark's completion (Clark, 1977). We define an order inspired by the program structure that is used to prioritize variable initializations in the DPLL algorithm. Since non-descendants of the query in $\mathcal{G}$ do not influence the query, they all get lowest priority.

**Definition 5.** For a query $q$ we define a partial order $\leq_q$ on clauses in a logical formula $\phi$ of a program as follows. Notice that every clause encodes (part of) exactly one rule from the program. Call $h(c)$ the head of the rule that clause $c$ encodes. Then if $h(c_1)$ and $h(c_2)$ are both descendants of $q$ in $\mathcal{G}$ we say that $c_1 \leq_q c_2$ if and only if $c_2$ is a descendant of $c_1$ in $\mathcal{G}$. If $h(c_1)$ is a descendant of $q$ but $h(c_2)$ is not, then always $c_1 \leq_q c_2$. This is similar to the stratification commonly used in logic programming (Apt et al., 1988).

The algorithm is initially called as EXPLAIN$(\phi, q, \emptyset, \leq_q, u)$ and returns a (partial) explanation of $q$. This explanation is subsequently completed. If the algorithm returns $\emptyset$, then a conflict was encountered during execution. App. C shows different ways to handle conflicts and perform the completion. The pseudo-code of EXPLAIN is given in Algorithm 1. Note that EXPLAIN enjoys the properties summarized in the following proposition.

**Proposition 3.1** (Properties)**.** EXPLAIN always terminates either by detecting a conflict or by returning a possible explanation. Additionally, the returned explanation $\nu$ is *parsimonious*. That is, the query $q$ is entailed in any completion of $\nu$.

*Proof.* Regarding termination, we first assume that no conflict can occur. Under this assumption we observe that each call of EXPLAIN visits a new node in $\mathcal{G}$ by following the corresponding topological

order. Since the graph is acyclic and finite, the algorithm eventually reaches the leaves or probabilistic facts, thus terminating. Now, if conflicts occur, EXPLAIN can terminate before reaching the leaves. Regarding parsimony, note that the algorithm initializes the minimal number of atoms at each rule to ensure that the query is always entailed. Therefore, remaining atoms do not have any impact on the query. □

The principle of parsimony introduced by EXPLAIN is beneficial for reducing the number of conflicts when searching for an explanation. App. D contains an analysis of how parsimony impacts the number of conflicts.

**Diversified sampling.** Now, we turn our focus to estimate $\Omega_q$ using EXPLAIN and show how we can adapt the value function $u$ to improve diversity. The key idea is to formulate $T$ calls to EXPLAIN as a deterministic Markov Decision Process (MDP) and leverage a recent result from the theory of GFlowNets (Bengio et al., 2021a). Let's introduce the elements of the MDP, namely its state, action space, transition and reward functions.

**Definition 6.** Given a set $\mathcal{V}$ containing all possible assignments, we define the *state space* of the MDP as $\mathcal{S} = \mathcal{V}^T$ with $T$ the number of samples outputted by EXPLAIN. We define the *initial state* of the MDP as $s_0 = [\nu_q, \emptyset, \ldots, \emptyset]$ with $\nu_q : \{q\} \to \{0, 1\}$ such that $\nu_q(q) = 1$. A state $[\mu_1, \ldots, \mu_T] \in \mathcal{S}$ is a *terminal state* when all $\mu_i$ are explanations of $q$, being the output of $T$ calls to EXPLAIN.

**Definition 7.** We characterize the *action space* using a mapping $\mathcal{A} : \mathcal{S} \to 2^{\mathcal{H}}$ that decides the pool of atoms for assignment, given the current state $s = [\mu_1, \ldots, \mu_{t-1}, \nu, \emptyset, \ldots, \emptyset]$, namely

$$\mathcal{A}(s) = \begin{cases} \mathcal{H} \setminus \mathrm{dom}(\nu) & \text{if } \nu \text{ is an explanation and } \mathrm{dom}(\nu) \neq \mathcal{H} \\ \mathcal{U} & \text{otherwise} \end{cases}$$

with $\mathcal{U}$ the unexplained atoms of $\nu$. So if $\nu$ is an explanation, we complete it and otherwise choose an atom to be explained.

**Definition 8.** Given a state $s = [\mu_1, \ldots, \mu_{t-1}, \nu, \emptyset, \ldots, \emptyset]$ and an action $a \in \mathcal{A}(s)$, the *transition function* of the MDP is defined as

$$\mathcal{T}(s, a) = \begin{cases} [\mu_1, \ldots, \mu_t, \nu_q, \emptyset, \ldots, \emptyset] & \text{if } \nu \text{ is a complete assignment} \\ [\mu_1, \ldots, \mu_{t-1}, \nu', \emptyset, \ldots, \emptyset] & \text{otherwise} \end{cases}$$

Specifically, $\nu' = \nu + a$ or $\nu' = \nu + \neg a$ with equal probability when $\nu$ is an explanation. Otherwise, if $\nu(a) = 1$ then $\nu' = \nu + \sum_b b$ with $b$ the literals in the body of the rule defining $a$ and if $\nu(a) = 0$ then $\nu' = \nu + \neg b$ for a random body literal $b$. The idea is that $\nu'$ either adds an explanation to or completes $\nu$. If $\nu$ is a complete explanation, we start with a new sample $\nu_q$.

We can now introduce the notion of diversity and the reward function of the MDP.

**Definition 9.** Given a terminal state $s_T = [\mu_1, \ldots, \mu_T]$, the diversity of $s_T$ is defined by $\delta(s_T) = |\{\mu_t \mid t = 1, \ldots, T\}|$. In other words, the diversity criterion counts the number of unique complete assignments in the state. The reward is $R(s) = \delta(s)$ for terminal states and $R(s) = 0$ otherwise.

**Definition 10.** The *generalized value function* $u : \mathcal{S} \times \mathcal{S} \to \mathbb{R}^+$ assigns values to transitions $s \to s'$.

These definitions lead us to the following result.

**Proposition 3.2** (Diversity guarantee). Given an MDP $(\mathcal{S}, \mathcal{A}, \mathcal{T}, R)$ and a generalized value function that is a solution to the flow equation for all states $s \in \mathcal{S}$

$$\sum_{s', a : \mathcal{T}(s', a) = s} u(s', s) = R(s) + \sum_{a \in \mathcal{A}(s)} u(s, \mathcal{T}(s, a)) \tag{2}$$

if we define the policy function as $\pi(a \mid s) = u(s, \mathcal{T}(s, a)) / \sum_{a' \in \mathcal{A}(s)} u(s, \mathcal{T}(s, a'))$ we have that

1. The transition graph associated to the MDP, i.e. nodes are states and edges are transitions, when starting from the state $s_0$ is a DAG.

2. The probability to reach a terminal state $s_T$ starting at $s_0$ following $\pi$ is $R(s_T) / \sum_{s'_T} R(s'_T)$.

*Proof.* For part 1, note that the MDP is constructed as to avoid cycles. Each transition either completes the corresponding assignment $\nu$ for sample $t$ in an incremental fashion (atoms are never

removed from the assignment) or starts a new sample $t + 1$ in case $\nu$ is already complete. Regarding part 2, the result follows directly from Proposition 2 in (Bengio et al., 2021a), because all assumptions are met. In particular, the graph is a DAG (from part 1). □

The proposition ensures that, by defining $u$ in a way to satisfy the flow equation, EXPLAIN can generate diverse samples, with probabilities proportional to the reward function $R$. However, defining such value function is in practice intractable, both computationally, as we need to solve a large system of linear equations, and in terms of storage, as we need to store the values of all transitions. Similarly to GFlowNets (Bengio et al., 2021a), we can avoid the intractability by parameterizing the function $u$ and learning it to minimize an objective quantifying the violation of the flow equation. We propose the following two simplifications for parameterizing the function $u$. First, let $u(s, s')$ depend only on the atom $a$ in $s$ that gets explained in the transition to $s'$. Second, we count how many times during the execution of EXPLAIN the edge from the head $a$ to the body literal $b$ is followed and call this number $n(a, b)$. All counts are initialized to $n(a, b) = 0$. We define $u = \alpha_{\text{nn}}(b) \gamma^{n(a,b)}$ with $\alpha_{\text{nn}}(b) = P(b \mid x)$ according to the neural network and $\gamma \in (0, 1]$ the damping factor. After sampling one explanation, the counts $n$ are incremented. Subsequently $u$ changes and the new value function is used in the next call to EXPLAIN. Because $0 < \gamma \leq 1$ whenever a transition is taken in one iteration, the probability of taking it again in the next iteration is dampened by $\gamma$. This encourages the algorithm to explore other explanations for the query and can thus lead to higher diversity. The $\alpha_{\text{nn}}$ factor is included so that the neural network can guide which edges to take. This leads to faster convergence. We provide the pseudo-code to train EXPLAIN for diversity in App. E.

## 3.2 AGREE: COMPUTE THE WEIGHT FUNCTION

We can now use the set of sampled models $\widetilde{\Omega}_q \subseteq \Omega_q$ to define the approximate posterior $\widetilde{Q}(f \mid x, q) = w_{x,f} \mathbb{1}_{f \in \widetilde{\Omega}_q}$ and to compute the KL objective in Eq. 1. This allows learning the weight function of the posterior $\widetilde{Q}$. By minimizing the KL with respect to the weight function we obtain an estimator of the true data log-likelihood. The estimator will be subsequently used as a training objective to update the perceptual component in the LEARN step. We can now provide the estimator and show its properties (the proofs are in App. F).

**Proposition 3.3** (Estimator). Assume $w_{x,f}^* = P(f \mid x) / \sum_{f' \in \widetilde{\Omega}_q} P(f'|x)$ and $\widetilde{Q}^*$ is the corresponding approximate posterior, then

1. $\widetilde{Q}^*$ is the global maximizer of Eq. 1.

2. The maximum of Eq. 1 is attained at $\sum_{j=1}^N \log \widetilde{P}_{x_j}(q_j)$, with $\widetilde{P}_x(q) = \sum_{f \in \widetilde{\Omega}_q} P(f \mid x)$.

3. For any $x$ and $q$ it holds that $\log \widetilde{P}_x(q) \leq \log P(q \mid x) \leq \log(1 - \widetilde{P}_x(\neg q))$.

The first result tells us that we can solve the maximization of Eq. 1 with respect to the weight function analytically, without requiring additional optimization steps. The value attained by the global maximizer is a biased estimator of the data log-likelihood in Eq. 1, used during the LEARN step. The third result states that the approximate objective provides bounds on the data log-likelihood commonly used during training by exact neural-symbolic methods. The bounds get tighter with more samples and become exact when all models of the query have been sampled at least once, i.e. $\widetilde{\Omega}_q = \Omega_q$. This convergence of bounds is also shown experimentally.

## 3.3 LEARN: LEARNING THE PERCEPTUAL COMPONENT

The LEARN step maximizes $\sum_{j=1}^N \log \sum_{f \in \widetilde{\Omega}_q} P(f|x)$ with respect to the parameters of the neural network $g$ using standard gradient descent strategies. Importantly, when $|\widetilde{\Omega}_q| = 1$, we observe that the loss is equivalent to the traditional negative cross-entropy loss. We can interpret the LEARN step as a standard supervised classification setting, where the labels are provided by the EXPLAIN and AGREE steps. See App. G for the pseudo-code.

## 4    RELATED WORK

We review recent works on neural-symbolic (NeSy) learning and scalable logical inference. For a more detailed discussion on NeSy, we refer the reader to recent surveys in Besold et al. (2021); Giunchiglia et al. (2022).

**NeSy relaxations**. Several NeSY systems address the scalability of inference with continuous relaxations based on fuzzy logic semantics (Badreddine et al., 2022; Giunchiglia & Lukasiewicz, 2021; Daniele & Serafini, 2019; Li & Srikumar, 2019; Gan et al., 2021; Sachan et al., 2018; Donadello & Serafini, 2019). However, these relaxations may introduce approximations that can yield different outputs for equivalent logic formulas (van Krieken et al., 2022; Grespan et al., 2021). In contrast, our work performs inference and learning without resorting to such relaxations while retaining a probabilistic interpretation.

**Neural inference strategies**. The work from Cornelio et al. (2023) proposes a neural-symbolic pipeline consisting of a symbolic engine and three neural modules. Specifically, a perception network mapping the images to their symbolic representations, a neural solver attempting to correct the symbolic representation, and a mask predictor to identify possible mistakes done by the neural solver. The symbolic solver then corrects the mistakes on the neural predictions. The neural modules are pre-trained under supervision and fine-tuned using reinforcement learning. In contrast, our simplified framework is based on a scalable logic engine and single neural component. Learning is performed by directly supervising the neural component using the sampled explanations using variational inference strategies. The work in van Krieken et al. (2023) introduces two neural modules for neural-symbolic learning: a perception component mapping input data to the probabilities of facts, and a neural reasoner component mapping probabilities to the query. Learning involves training the neural reasoner to mimic synthetic input/output pairs obtained through logical inference, and then training the perception component in a supervised manner using the frozen neural reasoner. In contrast, our work only requires a perception component and utilizes a sampling algorithm that guarantees logically consistent solutions while ensuring diversity. The Neural Theorem Prover (Minervini et al., 2020a;b) introduces a continuous and differentiable relaxation of the backward-chaining logic reasoning algorithm. In contrast, our approach does not rely on relaxations or differentiability through the logic program. Other neural sampling strategies, such as GFlowNets (Bengio et al., 2021b;a; Zhang et al., 2022), treat sampling as a sequential decision-making process and learn a policy based on a reward function. However, sampling with hard constraints and exploring solution modes from logical programs remains a challenging problem (Ermon et al., 2012; Sansone, 2022).

**Logical inference/sampling strategies**. Probabilistic logical inference can be performed exactly by transforming the logical program into a probabilistic circuit (PC) through knowledge compilation (Darwiche & Marquis, 2002). This allows for efficient evaluation in polynomial time (Choi et al., 2020; Xu et al., 2018; Ahmed et al., 2022; 2023). Alternatively, approximate strategies can be used to avoid the computational burden of knowledge compilation, but they may result in biased learning (Manhaeve et al., 2021; Huang et al., 2021; Skryagin et al., 2022) and in lacking guarantees on the uniformity/quality of the sampled solutions (Jerrum et al., 1986; Bellare et al., 2000). The assumption of uniform distribution of worlds is often made when sampling solutions from a CNF formula, and various uniform samplers have been proposed with theoretical guarantees on query complexity and uniformity Meel (2022). State-of-the-art samplers based on hashing-based methods and SAT solvers achieve approximate uniformity by partitioning the solution space into smaller regions (Eén & Sörensson, 2003; Soos et al., 2009; de Moura & Bjørner, 2008; Soos et al., 2020; 2009; Yang & Meel, 2021). However, existing samplers focus on sampling for CNF formulas without considering the structure of the logic program. In contrast, our algorithm focuses on a stronger criterion than uniformity, namely diversity, and on leveraging the logical structure of the program.

## 5    EXPERIMENTS

Experiments are provided to support our theoretical claims. We show that we can learn to generate diverse samples using the EXPLAIN algorithm. Then the importance of diversity and the AGREE step are illustrated by looking at the convergence of the learning objective. Lastly, we apply the EXAL paradigm to the MNIST addition learning task. More details on our implementation and a link to the code is in App. G. Our learning setup is explained in App. H.

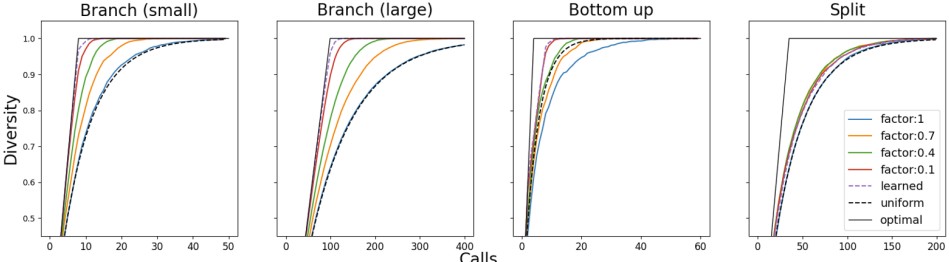

Figure 4: Diversity as a function of EXPLAIN calls for different programs and sampling strategies.

## 5.1 DIVERSITY (EXPLAIN STEP)

As a first experiment, we evaluate the diversity of different sampling strategies. Programs with a varied number of atoms, ranging between 15 and 60, have been generated to run the algorithm on. The branch program has a tree-like structure, forcing the EXPLAIN algorithm to make many choices in succession. The split program alternates between layers of conjunctions and disjunctions. The bottom up program is constructed starting at the facts and defining new atoms recursively using the atoms that already exist. We count how many unique samples are generated during subsequent calls to the algorithm, i.e., the diversity. The counts at each time step are averaged out over 200 reruns. This is done for fixed values of the factor $\gamma$, as well as for a $\gamma$ that is learned using the diversity optimizing algorithm in App. G. We set $\alpha_{\text{nn}} = 1$ for this experiment, since there is no neural component. As baselines, a uniform sampler with replacement is considered and also the optimal sampler that maximizes diversity at every time step. Figure 4 shows the evolution of the normalized diversity for each strategy.

Although the EXPLAIN algorithm is not guaranteed to sample uniformly, in many instances it is close to uniform. This can be seen by the overlapping of the uniform line and the line with $\gamma = 1$. For the bottom up program EXPLAIN is however not uniform and performs worse for $\gamma = 1$. The diversity of EXPLAIN is typically increased by choosing a smaller $\gamma$ and can get close to the optimal diversity. In some instances, such as the split program, having $\gamma < 1$ increases diversity, but there is little difference in diversity for the different values of $\gamma$. We also see that learning to optimize the diversity performs better than sampling uniformly.

## 5.2 CONVERGENCE OF BOUNDS (AGREE STEP)

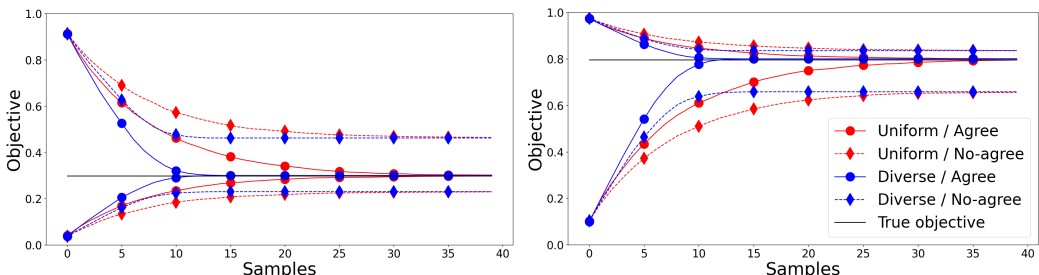

Figure 5: Convergence of objectives for increasing sample count, shown for two programs.

We now illustrate the importance of diversity and of the AGREE step. For this purpose we compare the objective function with the true likelihood of the query for different number of samples. More concretely, explanations are sampled for the query and for its negation, using both the diverse and non-diverse versions of EXPLAIN. Then the objectives with and without the AGREE step are calculated. The results for all four combinations are shown in Figure 5. The samples for the query give a lower bound, whereas samples for its negation give an upper bound. The AGREE step is necessary for the objectives to converge to the true likelihood. Furthermore, convergence happens faster when the samples are more diverse.

Table 1: Test accuracy of predicting the correct sum of two sequences of MNIST digits of length $N$. Accuracies of A-NeSI, DeepStochLog and Embed2Sym were taken as reported by van Krieken et al. (2023). EXL is our EXAL method without the AGREE step.

| Method | EXAL | A-NeSI | DeepStochLog | Embed2Sym | Reference |
|--------|------|--------|--------------|-----------|-----------|
| $N = 2$ | $94.96 \pm 0.42$ | $95.96 \pm 0.38$ | $96.40 \pm 0.10$ | $93.81 \pm \phantom{0}1.37$ | $96.06$ |
| $N = 4$ | $91.43 \pm 0.71$ | $92.56 \pm 0.79$ | $92.70 \pm 0.60$ | $91.65 \pm \phantom{0}0.57$ | $92.27$ |
| $N = 15$ | $71.60 \pm 3.30$ | $75.90 \pm 2.21$ | T/O | $60.46 \pm 20.36$ | $73.97$ |

## 5.3 SCALING MNIST ADDITION (EXPLAIN, AGREE, LEARN STEPS)

We evaluate our approach on the standard neural-symbolic experiment of learning to sum MNIST digits (Manhaeve et al., 2018; 2021). The input to this task is 2 sequences of $N$ MNIST images, where each sequence represents a decimal number of length $N$. The desired output is the sum of these two numbers, which is also the only supervision. The size of the possible assignments of values to each digit in each sequence ($10^{2N}$) in combination with the distant supervision is what makes learning to sum MNIST digits a challenging task.

Table 1 shows how we compare to the state of the art for $N = 2$, $N = 4$ and $N = 15$ digits. EXAL is competitive with A-NeSI and provides state-of-the-art performance, with the desired reference accuracy always within margin of error. This reference expresses the expected performance of predicting the correct sum of two $N$ digit numbers given a $99\%$ accurate digit classifier. Additionally, EXAL does not require finetuning of a neural approximation of the logic component or defining appropriate priors up front, in contrast to A-NeSI. Instead, it exploits the given program to directly acquire a fitting proposal distribution.

This direct proposal translates into a scalable and sample efficient overall procedure; the reported accuracies were reached after around 75, 28 and 78 minutes on average for $N = 2$, $N = 4$ and $N = 15$ respectively. A hold out-validation set was used to avoid overfitting, which was the cause for the relatively long duration of the $N = 2$ case as its validation accuracy remained on a long plateau. Comparable test accuracies can already be reached after just 25 minutes. Moreover, only 100 samples needed to be drawn for $N = 2$ and $N = 4$. For $N = 15$, 200 samples proved sufficient. Considering the exponentially growing size of the search space, these numbers are more than reasonable.

## 6 DISCUSSION & LIMITATIONS

There are two limitations on the programs that EXPLAIN can handle, namely being ground and acyclic. A first-order program can be grounded, but this can be computationally expensive. In the future we can look into a lifted version of the EXPLAIN, AGREE and LEARN paradigm. EXPLAIN will still terminate if the program contains cycles, but is not guaranteed to return an explanation. Also here it might be possible to alter the algorithm or look into different program semantics to correctly obtain explanations.

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
