# A    DERIVATION OF ELBO

The optimization of the log-likelihood can be framed in terms of a KL-divergence objective.

$$
\begin{aligned}
\log P(\mathcal{D}) \propto \sum_{j=1}^{N} \log P(q_j|x_j) &= \sum_{j=1}^{N} \log \sum_{f \in \{0,1\}^{|\mathcal{F}|}} P(q_j|f) P(f|x_j) \\
&= \sum_{j=1}^{N} \log \sum_{f \in \Omega_{q_j}} P(f|x_j) \\
&= \sum_{j=1}^{N} \log \sum_{f \in \Omega_{q_j}} Q(f|x_j, q_j) \frac{P(f|x_j)}{Q(f|x_j, q_j)} \\
&\geq \sum_{j=1}^{N} \sum_{f \in \Omega_{q_j}} Q(f|x_j, q_j) \log \frac{P(f|x_j)}{Q(f|x_j, q_j)} \\
&= -\sum_{j=1}^{N} \mathrm{KL}(Q_j, P_j)
\end{aligned}
$$

# B    SINGLY DEFINED TRANSFORMATION

To make a program $\mathcal{P}$ singly defined, the following transformation can be applied. Assume there is an atom $a$ that violates the singly defined property by occurring in the head of several rules $a \leftarrow b_{i,1}, ..., b_{i,m(i)}$ for $i \in \{1, ..., n\}$. These rules can be replaced by the rules $a \leftarrow \neg a'$ and $a' \leftarrow \neg b'_1, ..., \neg b'_n$ and $b'_i \leftarrow b_{i,1}, ..., b_{i,m(i)}$ where $a'$ is an auxiliary variable, to be interpreted as the negation of $a$. Call $\mathcal{P}'$ the program obtained by applying this replacement to all violating atoms. Note that $\mathcal{P}'$ is singly defined. The program $\mathcal{P}'$ has some auxiliary variables and thus a larger Herbrand base. Given a model $m$ of $\mathcal{P}'$, we say that its projected model is the intersection of $m$ and the Herbrand base of $\mathcal{P}$, i.e. we only consider the atoms that occurred originally in $\mathcal{P}$.

**Proposition B.1** (Equivalent Semantics). The models of $\mathcal{P}$ and the projected models of $\mathcal{P}'$ are the same set.

*Proof.* Let $m$ be a model of $\mathcal{P}$, then consider two cases. Either $a \in m$ or $a \notin m$. If $a \in m$, then for at least one $i$ all $b_{i,j}$ have to be true according to the semantics of $\mathcal{P}$. This means that $b'_i$ is true and hence $a'$ is false. Thus the replacement rules in $\mathcal{P}'$ are satisfied. If $a \notin m$, then for all $i$ at least one $b_{i,j}$ is false. As a result all $b'_i$ are false and $a'$ is true. Again all replacement rules in $\mathcal{P}'$ are satisfied. In any case $m$ is also a model of $\mathcal{P}'$.

Conversely, let $m$ be a projected model of $\mathcal{P}'$ and consider again two cases. If $a \in m$, then $a'$ is false according to $\mathcal{P}'$ and thus there is some $i$ for which $b'_i$ is true. This in turn means that for this $i$ all $b_{i,j}$ are true. Hence the body of the $i$th rule in $\mathcal{P}$ is true and all rules in $\mathcal{P}$ are satisfied. If $a \notin m$, then $a'$ is true and all $b'_i$ are false. For every $i$ at least one $b_{i,j}$ is false and thus none of the rules in $\mathcal{P}$ have a true body. This satisfies $a$ being false in $\mathcal{P}$. Any model of $\mathcal{P}'$ is thus a model of $\mathcal{P}$.    □

# C    CONFLICT HANDLING AND COMPLETION

During the execution of EXPLAIN (Algorithm 1) it is possible to encounter a conflict. This happens when the algorithm wants to assign a value $v$ to an atom $a$ that is different from the value that the assignment $\nu$ has already assigned to $a$. Note that the execution of EXPLAIN is probabilistic. It is thus possible that some probabilistic choices lead to a conflict whereas others do not. There are several policies for dealing with conflicts. The easiest policy is to simply stop execution and restart the EXPLAIN algorithm from the top. Another policy is to backtrack to a probabilistic choice and choose a different option. Also here there are several options. The algorithm can backtrack to the

---

**Algorithm 2** Pre-training algorithm PRE-TRAIN$(\mathcal{D}, u, \eta)$

---

1: **Input:** Dataset $\mathcal{D}$, generalized value function $u$ and learning rate $\eta$.
2: **Output:** optimized generalized value function.
3: **while** not converged **do**
4:     Compute

$$J(u; \mathcal{D}) = \sum_{t=1}^{KT} \left\{ \log \left[ \frac{\epsilon + \sum_{s,a:\mathcal{T}(s,a)=s_t} u(s, s_t)}{\epsilon + \mathcal{R}(s_t) + \sum_{a' \in \mathcal{A}(s_t)} u(s_t, \mathcal{T}(s_t, a'))} \right] \right\}^2$$

5:     Update $\gamma \leftarrow \gamma - \eta \nabla_\gamma J(u; \mathcal{D})$
6: **end while**
7: **return** $u$

---

nearest probabilistic choice, or backtrack up to the point where the initial assignment for $a$ was made. It is also possible to set a limit on the amount of backtracking that EXPLAIN can do before it decides to restart. Although Algorithm 1 stops execution upon encountering a conflict, in our implementation we employ backtracking to the nearest probabilistic choice.

If EXPLAIN does not encounter a conflict, it returns an explanation of the query $q$. This explanation is however not necessarily a complete assignment. In principle one can choose any distribution to complete these partial assignment, but there are two distributions that are particularly useful in this context. The first is to set the value of $c$ to true with probability $P(c = 1|x)$ for every $c$ that has not yet been assigned, i.e. to use the probabilities of the neural network. The second is to complete the partial assignment uniformly. We use the first option for experiments where a neural network is available and the second option otherwise.

## D PARSIMONY AND CONFLICTS

| | P20 | P50 | P100 |
|---|---|---|---|
| Parsimony | $0.04 \pm 0.26$ | $3.91 \pm 3.60$ | $1.82 \pm 2.29$ |
| No Parsimony | $23.12 \pm 13.80$ | $74.19 \pm 36.59$ | $379.92 \pm 716.36$ |

Table 2: Conflict counts for different programs when executing EXPLAIN parsimoniously or not.

EXPLAIN (Algorithm 1) propagates assignments in a parsimonious way, i.e. for every clause it locally assigns the minimum number of variables to make the clause true instead of assigning all variables in the clause. Parsimony lowers the number of conflicts that EXPLAIN encounters during execution. This is empirically observed in programs with 20, 50 and 100 atoms. The programs are generated by recursively generating rules for the atoms, which have an average in-degree of 3.0 and out-degree of 1.9. The number of conflicts during the execution of EXPLAIN are counted and averaged over 10 000 runs. Table 2 shows the results with and without parsimonious assignments.

## E DIVERSIFIED SAMPLING

In order to pre-train the sampling algorithm, we first sample $K$ models using EXPLAIN with parameter $\gamma = 1$. This is equivalent to make uniform decisions at each choice node. This gives us a dataset $\mathcal{D} = \{(s_{t-1}, s_t, \mathcal{R}(s_t))\}_{t=1}^{KT}$, where $T$ is the size of the sample set to compute the diversity criterion. Importantly, we are using the same objective criterion proposed in Bengio et al. (2021a). More details are provided in Algorithm 2.

## F PROOF OF PROPOSITION 3.3

**Proposition F.1** ([Restated 3.3]. Estimator and Properties]Assume $w_{x,f}^* = P(f|x) / \sum_{f' \in \widetilde{\Omega}_q} P(f'|x)$ and $\widetilde{Q}^*$ is the corresponding approximate posterior, then

    1. $\widetilde{Q}^*$ is the global maximizer of Eq. 2.

2. The global of Eq. 2 is attained at $\sum_{j=1}^{N} \log \widetilde{P}_{x_j}(q_j)$, where $\widetilde{P}_x(q) = \sum_{f' \in \widetilde{\Omega}_q} P(f'|x)$.

3. For any data point $x \in \mathcal{R}^D$ and Boolean query $q$, we have that $\log \widetilde{P}_x(q) \leq \log P(q|x) \leq \log(1 - \widetilde{P}_x(\neg q))$.

*Proof.* **(Part 1)** Without loss of generality, let's consider $w_{x,f} = \frac{\widetilde{w}_{x,f}}{\sum_{f' \in \widetilde{\Omega}_q} \widetilde{w}_{x,f'}}$. Eq. 2 can be therefore rewritten in the following way:

$$\log P(\mathcal{D}) \geq \sum_{j=1}^{N} \sum_{f \in \Omega_{q_j}} \widetilde{Q}(f|x_j, q_j) \log \frac{P(f|x_j)}{\widetilde{Q}(f|x_j, q_j)}$$

$$= \sum_{j=1}^{N} \sum_{f \in \widetilde{\Omega}_{q_j}} w_{x_j,f} \log \frac{P(f|x_j)}{w_{x_j,f}}$$

$$= \sum_{j=1}^{N} \sum_{f \in \widetilde{\Omega}_{q_j}} \frac{\widetilde{w}_{x_j,f}}{\sum_{f' \in \widetilde{\Omega}_{q_j}} \widetilde{w}_{x_j,f'}} \log \frac{P(f|x_j) \sum_{f' \in \widetilde{\Omega}_{q_j}} \widetilde{w}_{x_j,f'}}{\widetilde{w}_{x_j,f}}$$

$$= \sum_{j=1}^{N} \left\{ \frac{1}{\sum_{f' \in \widetilde{\Omega}_{q_j}} \widetilde{w}_{x_j,f'}} \sum_{f \in \widetilde{\Omega}_{q_j}} \widetilde{w}_{x_j,f} \log \frac{P(f|x_j)}{\widetilde{w}_{x_j,f}} + \log \sum_{f \in \widetilde{\Omega}_{q_j}} \widetilde{w}_{x_j,f} \right\}$$

$$\doteq \mathcal{L}(\{\widetilde{w}_{x_j,f}\}_{j=1,f}^{N})$$

Now, by computing $\nabla_{\widetilde{w}_{x_i,\xi}} L(\{\widetilde{w}_{x_j,f}\}_{j=1,f}^{N})$ and equating it to 0, we obtain the following equation for all $j \in \{1, \ldots, M\}$:

$$\sum_{f \in \widetilde{\Omega}_{q_j}} \widetilde{w}_{x_j,f} \log \frac{P(f|x_j)}{\widetilde{w}_{x_j,f}} = \sum_{f \in \widetilde{\Omega}_{q_j}} \widetilde{w}_{x_j,f}$$

$$\sum_{f \in \widetilde{\Omega}_{q_j}} \widetilde{w}_{x_j,f} \left\{ \log \frac{P(f|x_j)}{\widetilde{w}_{x_j,f}} - 1 \right\} = 0$$

Since we are dealing with probabilities (non-negative quantities) the above equality can be satisfied only in the case $\widetilde{w}_{x_j,f} = P(f|x_j)$ for all $f$ and $j$. Additionally, since Eq. 2 is as sum of concave functions, the resulting $w_{x,f}^*$ is a global maximizer.

**(Part 2)** We can substitute the result in part 1 into $\mathcal{L}(\{\widetilde{w}_{x_j,f}\}_{j=1,f}^{N})$, that is:

$$\mathcal{L}(\{P(f|x_j)\}_{j=1,f}^{N}) = \sum_{j=1}^{N} \log \sum_{f' \in \widetilde{\Omega}_{q_j}} P(f'|x_j)$$

and by defining $\widetilde{P}_{x_j}(q_j) = \sum_{f' \in \widetilde{\Omega}_{q_j}} P(f'|x_j)$, we obtain

$$\mathcal{L}(\{P(f|x_j)\}_{j=1,f}^{N}) = \sum_{j=1}^{N} \log \widetilde{P}_{x_j}(q_j)$$

**(Part 3)** We observe that the predictive log-likelihood for any single data point $(x, q)$ is given by the following expression:

$$\log P(q|x) = \log \sum_{f \in \{0,1\}^M} P(q|f, x) P(f|x)$$

$$= \log \sum_{f \in \{0,1\}^M} \mathbb{1}[f \models q] P(f|x)$$

$$= \log \sum_{f \in \Omega_q} P(f|x)$$

---

**Algorithm 3** Algorithm performing one training iteration TRAIN$(\mathcal{D}, u, \eta, \mathcal{G})$

---

1: **Input:** Dataset $\mathcal{D}$, learning rate $\eta$, value function $u$, DAG $\mathcal{G}$.
2: **Output:** updated $\theta$.
3: Get batch of samples $\{(x_j, q_j)\}_{j=1}^N \leftarrow \mathcal{D}$
4: **while** for all $j \in \{1, \ldots, N\}$ **do**                                     ▷ EXPLAIN
5:     $\widetilde{\Omega}_{q_j} \leftarrow \text{EXPLAIN}(q_j, 1, \nu_{q_j}, u, \mathcal{G})$
6: **end while**
7: Compute $\mathcal{K}(\theta) \leftarrow \sum_{j=1}^N \log \sum_{f \in \widetilde{\Omega}_{q_j}} P(f|x_j)$              ▷ AGREE
8: Update $\theta \leftarrow \theta + \eta \nabla_\theta \mathcal{K}(\theta)$                            ▷ LEARN
9: **return** $\theta$

---

By part 2, we know that the following inequality holds (as $\widetilde{\Omega}_q \subseteq \Omega_q$):

$$\log \widetilde{P}_x(q) = \log \sum_{f \in \widetilde{\Omega}_q} P(f|x) \leq \log \sum_{f \in \Omega_q} P(f|x) = \log P(q|x)$$

One can easily prove that:

$$\log P(q|x) \leq \log(1 - \widetilde{P}_x(\neg q))$$

$\square$

## G  OVERALL TRAINING ALGORITHM

We provide the pseudocode of the EXPLAIN, AGREE and LEARN paradigm in Algorithm 3. Our implementation can be found at `https://anonymous.4open.science/r/exal-AE49`.

## H  DETAILS ABOUT EXPERIMENTS

### H.1  DIVERSITY

The diversity experiment has been performed on 9 programs, of which 4 are shown in Figure 4. These are 3 branch, 3 bottom-up and 3 split programs. The branching factor of the branch programs are 3, 3 and 10 and have a depth of 3, 5 and 3 respectively. The bottom-up programs have as parameters the fraction of facts, which is always set to 0.5, the number of atoms, respectively 20, 60 and 60, and the in-degree, respectively 3, 3 and 4. The split programs alternate between conjunction and disjunctions for the atom definitions. All 3 programs have a depth of 4 and a conjunction size of 3 whereas the disjunction size varies from 2 to 4 inclusive.

### H.2  CONVERGENCE OF BOUNDS

To observe the convergence of bounds, 3 programs have been created, each with 24 atoms. The programs are created so that exactly half of the models make the query true. The probabilities of the facts are then varied in order to set the probability of the query to 0.3, 0.8 or 0.5, of which the first two are shown in Figure 5. For the diverse sampling algorithm we have used EXPLAIN with a factor of $\gamma = 0.1$.

### H.3  SCALING MNIST ADDITION

**Dataset.**     Generating the data for the MNIST addition experiment Manhaeve et al. (2018) on two sequences of $N$ digits is a straightforward process. It involves randomly selecting $2N$ images from the MNIST dataset and concatenating them to create two distinct sequences, each with a length of $N$. To supervise these sequences, we easily obtain the desired values by multiplying the labels of the selected MNIST images by the appropriate power of 10. We then sum the resulting sequence of values for each number and further sum the two resulting numbers. It is important to note that each MNIST image is only allowed to appear once in the sequences. Hence, the dataset consists of $\lfloor 60000/2N \rfloor$ sequences available for learning. The test set follows a similar procedure, using the test set partition of the MNIST dataset.

**Modelling.**     In this experiment, a traditional LeNet LeCun et al. (1998) neural network is utilized. The network architecture consists of two convolutional layers with 6 and 16 filters of size 5, employing ReLU activations. These layers are followed by a flattening operation. Subsequently, three dense layers with sizes of 120, 84, and 10 are employed. The first two dense layers also utilize ReLU activations, while the final layer applies a softmax activation. The network outputs the probabilities indicating the likelihood that each image in the two sequences corresponds to a specific digit. The logic program for MNIST addition is

$nn(Img, 0) :: digit(Img, 0); ...; nn(Img, 9) :: digit(Img, 9)$.

$number([], 0)$.

$number([Img|Rest], N) \leftarrow digit(Img, D), number(Rest, R), N$ is $D + 10 * R$.

$sum(Imgs1, Imgs2, S) \leftarrow number(Imgs1, N1), number(Imgs2, N2), S$ is $N1 + N2$.

where $nn(Img, D)$ is the probability that $Img$ represents digit $D$ according to the neural network.

**Hyperparameters.**     For this experiment, we adopted the standard Adam optimizer with a learning rate of $10^{-3}$, known for its reliable performance. Other critical hyperparameters include the number of samples drawn by EXAL and the number of epochs for training, both of which vary based on the value of $N$. When learning to sum 2 and 4 digits, we drew 100 samples and trained for maximum 100 epochs. For the case of $N = 15$, these number of samples was set to 200 samples.