# OpenReview forum: "EXPLAIN, AGREE and LEARN: A Recipe for Scalable Neural-Symbolic Learning"
_ICLR.cc/2024/Conference — ICLR 2024 Conference Withdrawn Submission_

### Official Review · Reviewer_Pp9a · 2023-10-27

**Soundness:** 3 good
**Presentation:** 2 fair
**Contribution:** 2 fair
**Rating:** 5
**Confidence:** 2

**Summary:**

The paper proposes the EXAL framework, which performs the 3 steps of EXPLAIN, AGREE, and LEARN to address scalable neuro-symbolic reasoning and learning.
EXAL produces Directed Acyclic Graphs (DAGs) given logic programs and performs sampling to achieve scalable probabilistic ne-sy reasoning and learning. Learning is performed by maximizing the log-likelihood.
The experiments have been conducted on a synthetic dataset and MNIST-Addition dataset.

**Strengths:**

The paper addresses an important topic in the neuro-symbolic community, i.e. the scalability issue, since the existing neuro-symbolic frameworks typically suffer from the large number of expressions to deal with practical problems. Thus, the community would benefit from the proposed approach to mitigate this issue. The method is presented with a certain formality.

**Weaknesses:**

Although I appreciate the paper’s aim, I found some concerns about the paper.


The main claim of the paper, i.e. the proposed framework is scalable, is not well-supported by the experiments for the following reasons.

### Lack of baselines that address the scalability issue in neuro-symbolic learning.
As cited in the paper, Scallop [1] addressed the scalability issue of DeepProbLog by introducing k-top operation, and NeurASP [2] and SLASH [3] addressed it by combining Answer Set Programming.
In my opinion, either of the following needs to be shown:
- The proposed framework is more scalable than the existing frameworks that address the scalability issue.
- The proposed framework gains additional advantages against existing frameworks. This needs to be empirically demonstrated. In the related work section, it is noted “.. they (DeepProbLog, Scallop, SLASH) may result in biased learning.”. If the proposed method can overcome this limitation empirically, outperforming the existing ones, it would strengthen the paper’s claim. Otherwise, it isn't easy to see how the proposed framework is superior to these existing ones.

### Lack of experiments on more challenging datasets/tasks.
Moreover, the experiments are conducted on synthetic datasets and MNIST-addition datasets, which are relatively simple. Empirical results on more complex/large datasets would be needed to support the main claim of the paper, i.e. the proposed method is scalable. Visual Question Answering with Reasoning  (VQAR) task has been proposed and addressed by Scallop. How would the proposed method scale to such a more challenging task? Would they gain advantages compared to existing methods such as Scallop and SLASH?


To this end, unfortunately, I found the paper somewhat not easy to follow.
- Many concepts are introduced and defined without clear specification of their motivations. Before/after introducing new formal concepts, explaining why we need them or providing more concrete examples would help readers understand the paper clearly.
- Some undefined terms are used. In definition 1, it is noted “rules in R are violated”, but the term“violated” is not defined.
- Some definitions of first-order logic are unclear. In Section 2, Herbrand base is explained as the set of all atoms, but Herbrand base usually refers to a set of *ground* atoms. Does $\mathcal{H}$ contain atoms with variables or only ground atoms?
- In Section 5.2, the proposed method is compared to different baselines, but no description of baselines is available in the main text. There are no references to the corresponding papers. The experimental setup needs to be clarified more to be self-contained.

I list some minor comments:
- In section 3.3, in the first line, the summation has $j$ as the index, but it is not used in the log-likelihood.
- $\nu$ seems dependent on ruleset R, i.e. when $\nu$ appears, a ruleset is assumed. Clarifying this more would help readers to understand.
- Providing some examples after definitions would be helpful.

**Questions:**

How would the proposed method scale to more challenging tasks such as VQAR? Would they gain advantages compared to existing ne-sy methods addressing the scalability issue such as Scallop and SLASH?

---

### Official Review · Reviewer_GwsC · 2023-10-30

**Soundness:** 2 fair
**Presentation:** 2 fair
**Contribution:** 2 fair
**Rating:** 3
**Confidence:** 4

**Summary:**

The paper proposes a probabilistic neural-symbolic framework called EXPLAIN, AGREE, and LEARN, which addresses the scalability of NeSy problems. The approach leverages diverse sampling by GFlowNets to obtain a set of possible explanations for the symbolic component and trains the neural component by approximating the true data log-likelihood based on the samples. Experiments show that all three stages are important to learning, and the proposed method is competitive with existing exact probabilistic NeSy approaches.

**Strengths:**

- The proposed three stages (EXPLAIN, AGREE, LEARN) for NeSy Learning are well-motivated.
- Experiments show that all stages are crucial and effective.

**Weaknesses:**

- The paper is missing a conclusion section, which would help in better understanding the contributions of the proposed method.
- Although the proposed methods based on sampling could be more efficient than exact inference, using sampling (GFlowNets) seems still not very efficient in addressing scalable NeSy problems in practice.
- The evaluation is relatively weak. The proposed method is only evaluated on the MNIST addition, which is quite toyish and limited. Also, it only compares with several exact probabilistic NeSy frameworks, which ignores some other approaches using approximate reasoning to address the scalability of NeSy problems (e.g., [1]). Moreover, it doesn't provide the time analysis of the proposed approach and its performance underperforms the recent baseline A-NeSI (which also uses GFlowNets).
- As pointed out by the authors, the proposed method is limited to certain programs, and it is hard to handle first-order programs.

[1] Scallop: From Probabilistic Deductive Databases to Scalable Differentiable Reasoning

**Questions:**

- Can the authors provide a brief analysis of the time complexity? Is the proposed framework more efficient than A-NESI?
- Is the method capable of scaling to relatively more complext tasks?

---

### Official Review · Reviewer_WFyQ · 2023-10-31

**Soundness:** 2 fair
**Presentation:** 2 fair
**Contribution:** 2 fair
**Rating:** 3
**Confidence:** 4

**Summary:**

The authors present a framework, termed Explain, Agree and Learn targeted at alleviating the scalability issues inherent in probabilistic NeSy learning. **Explain** leverages sampling to obtain *diverse* set of possible explanations encouraged by a newly-introduced diversity criterion. **Agree** assigns importance weights to the sampled explanation. Finally, **Learn** updates the neural component with direct supervision *without the need to backpropagate through the symbolic component*

**Strengths:**

- The overarching goal of the paper is a worthy one: NeSy approaches can indeed be very computationally expensive, and despite there being a plethora of proposed approximate approaches, it is still very much an active area of research.

**Weaknesses:**

- I found the paper quite hard to follow. The individual components, e.g. each definition on its own, are easy to grasp, but it's hard to keep track of how everything fits together. I feel that a running example would help tremendously.

- In the light of the questions I posed in the section below, in addition to the lacking experimental evidence on the MNIST-addition experiment, it's not clear to me that the paper's contributions are substantial enough, although I'd be open to changing my mind. The authors argue for the superiority of their approach on the basis of the the low sample complexity as well as training times, but do not provide any comparison in terms of timing for the (better performing) baselines.

**Questions:**

- In the background section, the penultimate line, am I correct in reading the expectation $\mathbb{E}_F[\mathbb{1}\_{F \models _\\mathcal{R}q}]$ as the expectation with respect to the distribution induced by the weights of the probabilistic facts of the indicator function? That is, when used as part of the expectation, $F$ is used in a probabilistic sense, and when used a part of the indicator function, $F$ is used in a logical sense? If so, I believe this should be made clear.

- I am confused by equation (1) and the preceding paragraph: How would one obtain the normalized weights $w_{x,f}$ without computing the weighted model count? Also, what are the $q_j$s in the dataset samples $\mathcal{D}$? Are they simply akin to targets, or ground-truth labels $y_j$ in standard fully-supervised learning?

- Regarding Section 3.1:
  - Can you please say more about Clark's Completion? Is it always possible to encode the ground program as a logical formula $\phi$? If so, what is the complexity of such an encoding?
  - A core contribution of yours seems to be the Explain algorithm, yet very little is said about how it work at an abstract level. Could you please state, intuitively, how it works? Could you also say more about your use of unit propagation (which is not refutation complete) and what consequences that has on your algorithm?
  - How does your Explain algorithm compare to top-k approximations of deepproblog or even sampling from the neural network's posterior [1,2]?

- How are the Agree and Learn step different from [2], where again, you simply maximize the approximate log-partition function, which in the case of [2] is approximated by sampling from the network's posterior distribution?

References:

[1] Manhaeve, R., Marra, G., De Raedt, L. (2021). Approximate Inference for Neural Probabilistic Logic Programming. In: Proceedings of the 18th International Conference on Principles of Knowledge Representation and Reasoning.

[2] Kareem Ahmed, Tao Li, Thy Ton, Quan Guo, Kai-Wei Chang, Parisa Kordjamshidi, Vivek Srikumar, Guy Van den Broeck, Sameer Singh. PYLON: A PyTorch Framework for Learning with Constraints. In AAAI 2021 & NeurIPS 2021.

---

### Official Review · Reviewer_fkWq · 2023-11-01

**Soundness:** 2 fair
**Presentation:** 2 fair
**Contribution:** 3 good
**Rating:** 5
**Confidence:** 4

**Summary:**

The paper introduces the EXAL paradigm (EXPLAIN, AGREE and LEARN) with the aim of addressing the scalability issues typical of Nesy learning. The method is tested on the MNIST addition task.

**Strengths:**

The paper proposes an interesting novel method that is of interest to the large AI community.

**Weaknesses:**

The paper could use a bit of rewriting in order to make it clearer, provide some more details into GFlowNets, be a bit more precise, and add all the relevant experimental analysis. Overall, it could be a very nice paper presenting some interesting ideas, but it needs a bit more work.

Below, more detailed comments can be found:

- The main claim of the paper is that this recipe helps in making NeSy models more scalable. However, there is no computational complexity analysis, nor there is an experiment that shows the gains in terms of memory/computational time with respect to other state-of-the-art models. Could you add something along these lines?

- The paper states that their goal is to address the scalability issues typical of Nesy learning. However, all their framework is based on ProbLog and its extensions. While ProbLog and DeepProbLog represent vital milestones for the NeSy community, conflating these models with all NeSy models is an overstatement. The title should probably then something about making ProbLog scalable, not all Nesy models. Similarly, the intro should be rewritten in this light.

- why did you define the weights $w_{x,f}$ such that $\sum_{f \in \Omega_q} w_{x,f}=1$? Does it have some probabilistic interpretation? If so, couldn't their sum be different from 1?

- when you give the definition of rules, you specify that the body is made of literals, what about $a$? (I guess it must always be positive, but it should be stated that it is an atom)

- the definition of explained seems the same as supported in logic programs. Why don't you draw a parallel between the two concepts?

- in definition 5 you suddenly start talking about clauses instead of rules. Why didn't you just stick to rules? Also, introducing clausing generates the problem that they have multiple rewritings (e.g., $\neg A \vee B$ can be rewritten both as $A \to B$ and $\neg B \to A$).

- Afterwards, you write that this is similar to the concept of stratification. How is it similar and how is it different? The definition of stratification revolves around the negative atoms appearing in the body of the rules, while in definition 5 the negation is not mentioned.

- in algorithm 1 there is the function "unit propagate" that I don't see explained anywhere (as this is an ML conference I think it should be explained). Again, in algorithm 1 there is written "smallest clause in $\phi$" according to $\le_q$. However, if I understand correctly, $\le_q$ defines a partial order. Hence, how do you define the "smallest clause"? If my understanding is correct, you might have multiple orders all valid, does picking different orders have an impact on the final explanation?

- as the paper seems to rely a lot on GFLowNets I would recommend the addition of a small paragraph where they are explained and the relevant theory is added

- why there is in caption of Table 1 written "EXL is our EXAL method without the AGREE step."? EXL is not mentioned anywhere else

- the authors state "EXAL is competitive with A-NeSI and provides state-of-the-art performance, with the desired reference accuracy always within margin of error". Could you please run a t-test to check the statistical significance of your claim?

**Questions:**

See above.

---

### Author Response · Authors · 2023-11-18
**Overall Considerations**

Dear Reviewers,

we would like to thank you for reading our paper and we appreciate your insightful comments.
Overall, we agree with most of the raised concerns. That is,
1. The paper can improve in terms of clarity (mentioned by almost all of you), we can indeed better relate the definitions introduced in the theory with the running example to let the readers appreciate the technicality and the depth of the theory, we can give more explanation to differentiate and valorize the sampling strategy over GFlowNets and top-k inference.
2. We can provide more evidence about the strengths of the proposed solution, by providing running time comparison for the experiments on MNIST addition and the baselines, so as to provide more concrete and convincing evidence about the scalability of the proposed approach.
3. We can include an additional non-MNIST experiment, thus showcasing the applicability of the approach to more difficult problems.

We have decided not to reply singularly to the reviews as the rebuttal period is not enough for us to seriously take into account these comments. Considering this, we have decided to withdraw the paper. However, we will take this action only after the rebuttal period to leave the time to read this message and leave the chance for any potential and additional comment to improve our paper.

Thank you again.

Sincerely,

The Authors